# A YOLOX-Based Automatic Monitoring Approach of Broken Wires in Prestressed Concrete Cylinder Pipe Using Fiber-Optic Distributed Acoustic Sensors

**DOI:** 10.3390/s23042090

**Published:** 2023-02-13

**Authors:** Baolong Ma, Ruizhen Gao, Jingjun Zhang, Xinmin Zhu

**Affiliations:** 1School of Water Conservancy and Hydroelectric Power, Hebei University of Engineering, Handan 056038, China; 2School of Mechanical and Equipment Engineering, Hebei University of Engineering, Handan 056038, China; 3Key Laboratory of Simulation and Regulation of Water Cycle in River Basin, China Institute of Water Resources and Hydropower Research (IWHR), Beijing 100038, China

**Keywords:** you-only-look-once (YOLO), deep learning, wire break monitoring, fiber-optic distributed acoustic sensor (DAS), prestressed concrete cylinder pipe (PCCP)

## Abstract

Wire breakage is a major factor in the failure of prestressed concrete cylinder pipes (PCCP). In the presented work, an automatic monitoring approach of broken wires in PCCP using fiber-optic distributed acoustic sensors (DAS) is investigated. The study designs a 1:1 prototype wire break monitoring experiment using a DN4000 mm PCCP buried underground in a simulated test environment. The test combines the collected wire break signals with the previously collected noise signals in the operating pipe and transforms them into a spectrogram as the wire break signal dataset. A deep learning-based target detection algorithm is developed to detect the occurrence of wire break events by extracting the spectrogram image features of wire break signals in the dataset. The results show that the recall, precision, *F*1 score, and false detection rate of the pruned model reach 100%, 100%, 1, and 0%, respectively; the video detection frame rate reaches 35 fps and the model size is only 732 KB. It can be seen that this method greatly simplifies the model without loss of precision, providing an effective method for the identification of PCCP wire break signals, while the lightweight model is more conducive to the embedded deployment of a PCCP wire break monitoring system.

## 1. Introduction

A prestressed concrete cylinder pipe (PCCP) consists of a thin-gauge steel cylinder that is either embedded in or lined with concrete, helically wrapped with high-tensile prestressing wire, and then coated with a dense mortar [1]. Because of its large capacity, high internal pressure, high soil coverage, and affordable price, PCCP is widely used in large-diameter water supply and drainage pipe networks [2,3,4]. Currently, over 35,000 km of PCCP has been installed in North America [5], and over 20,000 km has been installed in China [6]. However, PCCPs can be damaged due to overloading, material defects, non-standard production and construction, and environmental corrosion [7,8,9], and these damages cause the breakage of wires, separation of the mortar coating, cracking of the mortar coating or concrete core, and rupture of the pipe [3]. Although PCCP has a very low failure rate, a PCCP pipeline burst is accidental and catastrophic, with little or no advance warning. A pipe burst will not only disrupt the regional water supply, but also cause traffic, environmental, sanitation, and other public safety incidents, resulting in great economic and social harm. The strength of PCCPs depends on the high-strength prestressed steel wire wound around the core, which produces a uniform compressive prestress on the core to compensate for the stresses generated by internal pressure and external loads. Wire breakage is a major factor that results in the failure of PCCPs [6,7]. When there are a number of broken wires in the same area, or the number of broken wires in a pipe reaches a certain percentage, the strength of the pipeline is significantly reduced, eventually leading to a pipe burst.

In the past two decades, some researchers have studied the failure mechanism of PCCPs caused by broken wires by establishing a numerical model of the PCCP broken wire effect, and revealed the relationship between the number of broken wires and PCCP failure. The main methods used to evaluate the performance of PCCPs with broken wires include finite element analysis and prototype tests, which investigate the relationship between the number of broken wires, the location of broken wires, and the maximum bearing internal pressure [10,11]. Zarghamee et al. [12] took the steel wire fracture area as the core to simulate the broken wire of the steel wire in the symmetrical band-shaped rectangular area, and used the ABAQUS nonlinear finite element model, which included the nonlinear stress–strain relationship of concrete. The failure characteristics of different types of PCCPs, such as the strain of the core concrete, the propagation of cracks, and the yield deformation of the steel tube wire, are simulated under complex loading conditions. You and Gong [13] established a three-dimensional solid model of PCCP wire breakage. The model took into account the nonlinear behavior of materials, pipe–earth interaction, prestress loss, and the comprehensive effects of internal and external loads. The results show that the prestress loss caused by wire breakage will accelerate the PCCP’s destruction process. Hajali [8,14] investigated the effect of different wire breakage ratios and locations on the structural behavior and performance of PCCPs using numerical simulation. Hu et al. [15] conducted a prototype test on a PCCP with an inner diameter of 2.8 m to investigate the effect of wire breakage on the load-bearing performance of the PCCP and determined the influence range of wire breakage using the strain of the mortar.

Detecting the fracture of prestressed wires in PCCPs has become a popular research topic in recent years. Among the studies, acoustic emission and the electromagnetic eddy current are the main methods used to evaluate the fracture of prestressed wires [16]. Elfergani [17] investigated the early corrosion signal characteristics of prestressing wires using acoustic emission to promote the development of monitoring technology for detecting wire breakage in PCCPs. Holley et al. [1] proposed and developed an acoustic fiber-optic (AFO) monitoring method to detect wire breakage, which was tested and verified by conducting experiments on a PCCP pipeline. When the prestressed steel wire breaks, energy is released, which propagates in the water in the form of sound waves [18,19,20]. AFO technology uses water in the pipe as the medium to transmit sound waves to the sensor. The arrival time of the disconnection signal to the sensors spaced along the length of the pipe is recorded, and the location of the disconnection is then determined [9]. Monitoring methods based on fiber-optic sensors have a major advantage over conventional non-destructive techniques in that they are capable of remotely distributed condition monitoring [21]. Huang et al. [5] proposed a method for monitoring and locating wire breakage in PCCP pipelines using fiber Bragg grating sensors. Gallagher et al. [22] focused on incorporating real-time monitoring technology into aqueduct protection procedures via AFOs, verifying the accuracy of the disconnection detection results, and extending the remaining service life.

Fiber-optic distributed acoustic sensors (DAS) are one of the most attractive and promising fiber-optic sensing technologies of the last decade [23]. DAS can continuously detect acoustic signals and vibrations within tens of kilometers, with high sensitivity and a high update rate. DAS technology offers many advantages over competing technologies because it is easy to deploy, immune to electromagnetic interference, cost-effective, and does not require inline amplification or power supplies. Whether based on optical time domain reflectometry (OTDR) or optical frequency domain reflectometry (OFDR), DAS systems include interrogators and sensing cables [24].

Nowadays, DAS has become a versatile technology in many fields, such as real-time vehicle detection [25], traffic flow detection [26], subsurface seismic monitoring [27], volcanic event monitoring [28], and airplane flutter monitoring [29].

Li et al. [30] proposed a DAS technique for monitoring and identifying wire breaks in PCCPs under different conditions, such as corrosion and hydrogen embrittlement. The results of the study show that DAS has recognition accuracy for vibrations, especially broken wires, and can quickly and efficiently capture broken wires and noise at multiple locations in a variety of environments. The disconnection signals in different environments have both similarities and differences. From the time domain perspective, such as amplitude, duration, short-term zero-crossing rate, and short-term energy, disconnection and noise can be effectively distinguished. The experimental results show that under the same internal water pressure condition, the disconnection signal is unrelated to the external factors causing the disconnection, such as corrosion and hydrogen embrittlement.

The detection and location of the broken wires is achieved by the time difference between the vibration signal and the signal received by the sensor. The identification of the broken wires in PCCPs based on DAS technology requires efficient automatic processing of the original data. Recently, as highly automated computer vision detection techniques have been widely used in structural health monitoring [31,32], computer vision and convolutional neural network (CNN)-based environmental sound event recognition have also achieved a great degree of development. The CNN, as a representative algorithm for deep learning, is generally composed of an input layer, a convolutional layer, a pooling layer, a fully connected layer, and an output layer. The translation, scaling, and rotation of the image show high robustness. The convolution layer is composed of multiple filters. The filter refers to the convolution kernel. Differently sized convolution kernels can extract different feature information. The product can easily extract the underlying features of edges and curves, and the high-level convolution can easily extract abstract features. The main advantage of a CNN is that it can automatically detect important features without any human supervision. CNNs have extraordinary performance for classifying different sounds based on spectrograms. It distinguishes between masked noise and original sound in time or frequency. Piczak [33] applied a CNN to the task of ambient sound classification with a limited dataset and simple data augmentation, and achieved a similar level of performance to other feature learning methods. The most common deep learning-based sound classification method is to convert audio files into images and then use neural networks to process the images [34]. Sound classification usually uses images in the form of spectral images. Zhang et al. [35] used a CNN to identify the image-like features of environmental sounds and achieved more satisfactory recognition precision. Mushtaq et al. [36] proposed an effective method for classifying environmental sounds based on a CNN in the form of a spectrogram with meaningful data augmentation. In this method, Mel spectrograms are used to define features from audio clips in the form of spectrogram images. A spectrogram can be seen as a visible representation of the spectrum of the audio signal, and these images are further used to define the classification in the learned model. Boddapati et al. [34] presented a method for converting audio clips into spectrograms and implemented well-known transfer learning models (GoogleNet and AlexNet) for these spectrograms for image recognition tasks. Peng et al. [37] used a CNN to analyze different types of knock signals monitored by a DAS system and achieved 85% recognition precision. Jakkampudi et al. [38] developed a CNN to automatically detect footstep signals in ambient seismic recordings from urban DAS arrays. Huot and Biondi [39] used a CNN to automatically detect car-generated seismic signals with the objective of removing them from the seismic recordings.

YOLO is an acronym for the term “you only look once”. This is a CNN-based one-stage detector where the input image is sampled only once, hence the name [40]. Compared with traditional object detection methods, it has several advantages [41]: (1) YOLO is very fast; (2) YOLO performs global inference on images when making predictions; (3) YOLO learns generalizable representations of objects.

The YOLO algorithm is iterated continuously; each version has improved on the previous version in terms of speed and accuracy of detection. The YOLO series has now evolved from YOLO V1 [41] to YOLOX [42]. YOLOX is a newly released object detection model that is an improvement on the previous YOLO algorithm [40]. YOLOX [42] changed the YOLO detector so that it was anchor-free. Advanced detection techniques were implemented, i.e., a decoupled head and the leading label assignment strategy SimOTA to achieve the most advanced performance on various object detection models.

A study by Stork et al. [43] used YOLO V3 to detect microseismic signals collected by a DAS system with precision exceeding that of manual detection by 80%, with only a 2% false detection rate. Luo et al. [44] designed a lightweight detection network called G-YOLOX for vehicle type detection. It is suitable for practical applications with an embedded device. Considering the problems in engineering, Zhang et al. [45] selected a YOLOX algorithm model to develop a fast and high-precision X-ray image detection method for contraband, and a map value of 91.6% was achieved.

Transfer learning techniques, i.e., to transfer knowledge or information from one or more domains (source domain) to another (target domain), have been applied and studied in different machine learning problems. Transfer learning is a process of overcoming the transcendental learning paradigm and using the information and expertise gained for a particular task to solve other related problems [46]. Pre-trained transfer learning models are usually divided into two parts; the first is the convolutional basis part, which consists of the set of convolutional and pooling layers, for which the elementary purpose of dividing the model is to identify the features of the image. The second part consists of fully relevant layers, often called classifiers, and the main task of this part is to classify the image using the detected features, which are identified by the convolutional basis. This transfer learning model is based on CNNs, which are more accurate, deeper, and more efficient to train, and it contains a more concise connection between the input and output layers. Each layer is connected to the other layers in a feedforward manner, there is no need to train the model from the beginning, the model is highly capable of feature propagation and manipulation, and it also significantly reduces the dependence on the number of samples in the dataset. For example, if there is a neural network for the source task, it is possible to freeze (or, say, share) most of its layers and fine tune only the last few layers to generate the target network [47]. Since the failure rate of large-diameter PCCPs is lower than that of small-diameter pipes, there are less historical data available for analysis and, for pipelines in operation, it is not possible to deliberately damage them to obtain wire break signal data, which results in a very limited number of samples in the PCCP wire break signal dataset. At the same time, the environment in which individual PCCP pipelines, or even each section of a pipeline, are located, the structure of the pipeline, the internal pressure, the depth of burial, etc., can vary greatly, as well as the relationship between broken wires and failure. Their adaptation to other sites or other distributed fiber-optic sensing systems can require significant time and human resources, as some of the features of the processing data can be very different, and this is where transfer learning models are particularly important and effective. In this study, transfer learning migrates from a large and complex multi-classification model to a simple single-classification model. The shallow frozen weights are sampled more generically, leaving a large number of low-contributing features in the well-tuned model; to remove these redundancies, the pruning algorithm [48] is used to prune well-tuned models for transfer learning. As a compression method for neural networks, pruning algorithms essentially remove connections between neurons or entire neurons, channels, or filters from the trained network to reduce the model complexity and to prevent overfitting, and to reduce the model size for subsequent embedded deployment.

In order to identify PCCP wire break events accurately and to support the development of a subsequent PCCP wire break monitoring system, the overall objective of this study is to develop a wire break detection method with high recognition precision and a small model size, with the following specific objectives.

(1) In a 1:1 prototype wire break monitoring test, a manual cut is used to simulate a prestressed wire break event and the acoustic signal of the wire break is collected by a distributed fiber-optic acoustic sensing system.

(2) Combining the wire break acoustic signal with the noise signal collected in the operating pipeline previously, the spectrogram dataset of the simulated wire break signal is created by synchrosqueezed wavelet transform without denoising.

(3) We fine tune the YOLOXs object detection model via transfer learning, and further simplify the model using a pruning algorithm while ensuring precision.

## 2. Fundamentals and Methods

### 2.1. Acoustic Signal Processing for Wire Breaks

In order to obtain texturally clear images, this study uses the synchrosqueezed wavelet transform [49] (SWT) to transform a one-dimensional acoustic signal into a spectrogram with time on the horizontal axis, frequency on the vertical axis, and amplitude represented by color. The SWT is obtained by sharpening the continuous wavelet transform [50] (CWT), for which the CWT of a signal s is defined as
(1)$$W_{s}(a,\tau )=\int s(t)a^{-\frac{1}{2}}\bar{\psi (\frac{t-\tau }{a})}dt$$
where ψ(x) is the wavelet basis function used to extract the instantaneous frequency line to redistribute $W_{s}(a,\tau )$ to obtain a focused time–frequency image.

The signal s(t) of a purely harmonic wave can be expressed as
(2)s(t)=Acos(ωt)

For a wavelet basis function $\hat{\psi }(\xi )$ concentrated on the positive frequency axis and satisfying $\hat{\psi }(\xi )=0$, ξ<0. $W_{s}(a,\tau )$ can be rewritten by Plancherel’s theorem as
(3)$$\begin{array}{l} W_{s}(a,\tau )=\frac{1}{2\pi }\int \hat{s}(\xi )a^{\frac{1}{2}}\bar{\hat{\psi }(a\xi )}e^{i\tau \xi }d\xi \\ \;\;\;\;\;\;\;\;\;\;\;\;\;\;=\frac{A}{4\pi }\int \left[\delta (\xi -\omega )+\delta (\xi +\omega )\right]a^{\frac{1}{2}}\bar{\hat{\psi }(a\xi )}e^{i\tau \xi }d\xi \\ \;\;\;\;\;\;\;\;\;\;\;\;\;\;=\frac{A}{4\pi }a^{\frac{1}{2}}\bar{\hat{\psi }(a\omega )}e^{i\tau \omega } \end{array}$$

When $\hat{\psi }(\xi )$ is concentrated at $\xi =\omega_{0}$, $W_{s}(a,\tau )$ will be concentrated at $a=\omega_{0}/\omega$. However, $W_{s}(a,\tau )$ will spread out around the horizontal line $a=\omega_{0}/\omega$ in the time scale plane, blurring the time–frequency line. To solve this problem, for any point (a,τ) satisfying $W_{s}(a,\tau )\neq 0$, the instantaneous frequency $\omega_{s}\left(a,\tau \right)$ of signal s can be expressed as
(4)$$\omega_{s}(a,\tau )=-i\frac{1}{W_{s}(a,\tau )}\frac{\partial }{\partial \tau }W_{s}(a,\tau )$$

To achieve a time–scale (τ,a) to time–frequency $\left(\tau ,\omega_{s}(a,\tau )\right)$ mapping, the synchrosqueezed transformation of Equation (4) can be expressed as
(5)$$T_{s}(\omega_{\ell },\tau )=\frac{1}{\Delta \omega }\sum_{\frac{a_{k}}{\left|\omega (a_{k},\tau )-\omega_{\ell }\right|}\leq \frac{\Delta \omega }{2}}W_{s}(a_{k},\tau )a_{k}^{-\frac{3}{2}}{\left(\Delta a\right)}_{k}$$
where $a_{k}$ is the degree of discretization, $\Delta a_{k}=a_{k}-a_{k-1}$, $\omega_{\ell }$ is the synchrosqueezed center frequency, $\Delta \omega =\omega_{\ell }-\omega_{\ell -1}$, and the squeeze range is $\left[\omega_{\ell }-\frac{\Delta \omega }{2},\omega_{\ell }+\frac{\Delta \omega }{2}\right]$.

### 2.2. Neural Network Architecture

In this study, YOLOXs are chosen as detectors for PCCP broken wire signals. The YOLO series is a single-stage target detection model, which was proposed by Redmon et al. [41] in 2016 for finding a particular object or some specific objects in a picture or a video. Compared to other target detectors, the YOLO series is simply constructed and highly accurate. Most importantly, for the huge amounts of data generated by a DAS system, which requires faster processing speeds [43], the greatest advantage of the YOLO series can be highlighted. Currently, the YOLO series has evolved from YOLO V1 to YOLOX. In YOLOX, Ge et al. [42] chose to add Decoupled Head, Data Aug, Anchor Free, and SimOTA components to improve the network precision, convergence speed, and performance in preventing overfitting.

YOLOXs are the smallest standard network models in YOLOX in size. The architecture of the YOLOXs-based broken wire spectrogram detection network is shown in Figure 1.

The basic components are as follows:

1. The Focus module. This module performs a slicing operation on the image, taking a value for every other pixel in the image, thus obtaining four images, making the W, H information concentrated in the channel space and expanding the input channel four times. The input [Batch_size, 640, 640, 3] output gives [Batch_size, 320, 320, 12], as shown in Figure 2.

2. The CBS module. This module consists of a convolution layer, a Batch Normalization (BN) layer, and a SiLU activation function. The convolution layer is the core component of the CNN, which can extract features by input data through filters, and the data are calculated by convolution to obtain the feature image $X_{i}$:(6)$$X_{i}=X_{i-1}⊗W_{i}+b_{i}$$
where $X_{i}$ is the feature image of layer i, $X_{i-1}$ refers to the input of this layer, ⊗ indicates the convolution calculation, $W_{i}$ indicates the weight of the neurons in this layer, and $b_{i}$ is the bias.

The BN layer normalizes the input data to prevent data gradients from disappearing or exploding and to improve the generalization ability of the network [51], and the BN layer is calculated as follows:(7)$$X_{BNi}=\frac{\gamma (X_{i}-\mu )}{\sqrt{\sigma^{2}+\varepsilon }}+\beta$$
where μ and $\sigma^{2}$ are the mean and variance calculated on a batch, ε is a minimal constant that maintains numerical stability, γ is a scaling factor, and β is a shift factor.

The main role of the SiLU activation function is to complete the nonlinear transformation of the data, solving the problem that the linear model has insufficient ability in representation and classification:(8)$$SiLU({\hat{X}}_{i})=\frac{{\hat{X}}_{i}}{1+e^{-{\hat{X}}_{i}}}$$

3. Center and Scale Prediction (CSP) module. The CSP_N module enhances the learning ability of the CNN composed of a CBS module, residual unit, and concatenate function. The CSP2 module improves the network feature integration.

4. Spatial Pyramid Pooling (SPP) module. This module consists of a pooling layer, concatenate function, and CBS module. The pooling layer is also called downsampling. The feature images obtained by convolutional computation generally require a pooling layer to reduce the amount of data, and the pooling operation can effectively avoid overfitting:(9)$$X_{i}=\psi (X_{i-1})$$
where ψ(x) is the pooling operation; there are two common pooling criteria, max pooling or mean pooling, i.e., taking the maximum or average value of the corresponding region as the pooled element value. The max pooling operation is used here.

### 2.3. Pruning Algorithm for YOLOXs Model

To simplify the model and improve the detection efficiency, this study uses structural pruning to prune the well-tuned YOLOXs model. Structural pruning involves removing the entire block of weights within a given weight matrix, so that no problematic weight matrix of sparsely connected patterns is generated. The specific pruning process is chosen to iteratively prune and retrain the filter layer by layer [52], which takes more time to train more epochs, but is much more tractable and restores a higher degree of the original precision. The pruning process with a pruning rate of *x* applied from the *i*-th convolution layer is as follows:

1. For each filter $F_{i,j}$, calculate the sum of the absolute values of its kernel weights $s_{j}$.

2. Prune the filters with the smallest *x* × 100% of $s_{j}$ and their corresponding feature images, and also prune the kernels corresponding to the pruned feature images in the next convolution layer.

3. Create a new kernel matrix for layers *i* and *i* + 1, and, at the same time, copy the remaining weight parameters into the new model.

### 2.4. Fusing Convolution Layers with BN Layers

In this study, the inference of the model is accelerated by fusing the convolutional layer with the BN layer, where μ and $\sigma^{2}$ are recomputed for each batch during the training phase. However, in the testing phase, the previous μ and $\sigma^{2}$ are no longer used, instead using the exponentially shifted average $\hat{\mu }$ and ${\hat{\sigma }}^{2}$ during the training process, so the convolution and BN layers can be combined to increase the inference speed without affecting the detection precision; the computational flow is as follows.

For the test phase, the BN layer calculation Equation (7) can be rewritten as follows:(10)$$X_{CBNi}=\frac{\gamma X_{i}}{\sqrt{{\hat{\sigma }}^{2}+\varepsilon }}-\frac{\gamma \hat{\mu }}{\sqrt{{\hat{\sigma }}^{2}+\varepsilon }}+\beta$$

Combining the convolution calculation Equation (6) with the BN layer calculation Equation (10), we have
(11)$$\begin{array}{l} X_{CBNi}=\frac{\gamma (X_{i-1}⊗W_{i}+b_{i})}{\sqrt{{\hat{\sigma }}^{2}+\varepsilon }}-\frac{\gamma \hat{\mu }}{\sqrt{{\hat{\sigma }}^{2}+\varepsilon }}+\beta \\ \;\;\;\;\;\;\;\;\;\;\;=X_{i-1}⊗\frac{\gamma W_{i}}{\sqrt{{\hat{\sigma }}^{2}+\varepsilon }}+\frac{\gamma (b_{i}-\hat{\mu })}{\sqrt{{\hat{\sigma }}^{2}+\varepsilon }}+\beta \end{array}$$

It can be seen that the fused convolution and BN layers are linear operations, and we assume that
(12)$$\begin{array}{l} {W^{\prime }}_{i}=\frac{\gamma W_{i}}{\sqrt{{\hat{\sigma }}^{2}+\varepsilon }}\\ {b^{\prime }}_{i}=\frac{\gamma (b_{i}-\hat{\mu })}{\sqrt{{\hat{\sigma }}^{2}+\varepsilon }}+\beta \end{array}\}$$

Rewriting Equation (11) gives
(13)$$X_{CBNi}=X_{i-1}⊗{W^{\prime }}_{i}+{b^{\prime }}_{i}$$

## 3. Brief Test Summary

### 3.1. Wire Break Monitoring Test

In this test, a time domain DAS system with a 10 m spatial resolution was set up as shown in Figure 3. A laser with a narrow line width is used as the optical source of the system. The continuous-wave light from the laser is modulated by an acoustic optical modulator (AOM) to generate the pulses. The modulated pulses are amplified by the first erbium-doped fiber amplifier (EDFA) and then they are launched into the sensing fiber through an optical circulator (CIR). The Rayleigh scattering signal is amplified to obtain a better signal-to noise ratio by the second EDFA and injected into one port of a 3 × 3 coupler through another circulator. Two Faraday rotating mirrors (FRM) are respectively connected to two ports on the other side of the coupler with the optical path difference of 5 m. The final interference information output from the coupler is collected by three photodetectors (PD).

To obtain more realistic wire break signals under the same conditions as the actual operating PCCP pipeline, four DN4000 mm PCCPs are buried underground, building the main structure of the test environment, with a length of 20 m and an internal diameter of 4 m, as shown in Figure 4. By filling the pipe with water and pressurizing it, the test pipe conditions are made close to the actual conditions. The test sets up both DAS and hydrophones, and only the results of the DAS equipment are selected for the subsequent study, so the hydrophone part is not discussed any further.

Prior to the test, seal panels are installed at both ends of the PCCP and customized cable entry seals are installed in the opening on the side walls of the manhole; a pressure- and water-resistant armored communication fiber-optic backbone cable is placed from the control room to the opening, and the armoring will attenuate the acoustic signal reaching the fiber core, thus reducing the signal-to-noise ratio. However, armoring is necessary to protect the fiber-optic cable; therefore, the armoring method must be identical in all phases of the experiment to ensure that the signal can be accurately identified; then, polyurea is used to seal the gap between the fiber-optic cable and the entry device, adhering the sensing fiber-optic cable to the inner wall of the PCCP. We then splice the backbone fiber-optic cable and the acoustic fiber tail cable, splice the tail cable in the control room, and connect the DAS instrument; we close the discharge valve, seal the manhole, fill the pipe with water, and use a pressure pump to pressurize the pipe after filling it with water and keep the pressure stable. An electric pick and cutter are used to cut windows in the outer mortar layer of the pipe to expose the outer prestressed wire for wire cutting operations.

The test is carried out by cutting the prestressed wire manually. To ensure the safety of the test, the number of broken wires of each PCCP is 15. Photographs of the test section are shown in Figure 5. The wire break events are monitored by the DAS system, and the field data are saved and demodulated offline.

The data collecting system in this test is a DAS system and a laptop, the sampling interval is 0.05 ms, the collecting time is 12 s, the number of data at each corresponding break point position is 240,000, with continuous recording, and the typical break waveform graph is shown in Figure 6.

### 3.2. Network Training

#### 3.2.1. Training Platform

The training platform for this study uses the Windows 10 operating system. The hardware configuration is as follows: CPU: Inter(R) Core(TM) I9-12900K CPU @ 3.90 GHz; GPU: NVIDIA GeForce RTX 3080ti. The programming environment is PyCharm 2020 and the deep learning framework is PyTorch.

#### 3.2.2. Training Dataset

The dataset used in this study contains the synthetic signals of the wire break signals collected from the tests and the previously collected background noise of the operating large-pressure PCCP through the synchrosqueezed wavelet transform to obtain a spectrogram image. To enrich the dataset and avoid overfitting, the 60 wire break signals recorded from the tests are randomly combined with the operating pipeline background noise data and minor leakage working condition data in this study. The pipeline background noise data were collected in a pipeline buried to a depth of 3 m in an urban environment with human activity and vehicle movements, and typical combination processes are shown in Figure 7.

After expanding the dataset, the synthesized signals were transformed by SWT to obtain 600 spectrograms of the simulated wire break signals, as in Figure 8.

Among them, 500 simulated wire break signal spectrograms obtained in combination with wire break signals numbered 1–50 are used as training and validation sets. The produced images of the dataset keep only the middle part of the image containing the break features, and we cut the image size uniformly to 640 × 640 pixels, and then manually mark the location of the break features, as in Figure 9. The 100 spectrogram images obtained in combination with the wire break signals numbered 51–60 are made into 24 frames of 480p rolling video in mp4 format as the test set, with the total duration of the video remaining the same as the sum of the times on the *x*-axis of the images.

#### 3.2.3. Fine Tuning YOLOXs

To obtain a YOLOXs-based spectrogram detection model for wire breaks, the manually labelled training and validation sets described in Section 3.2.2 are used to fine tune the YOLOXs network by transfer learning, with the experiment setting the batch size to 8 and the number of training epochs to 200; the initial learning rate is 1 × 10^−4^ and the number of classes is 1. After 200 epochs, the model converged and the loss function for the whole training process is as shown in Figure 10.

#### 3.2.4. Pruning the YOLOXs Model

After iterative pruning and retraining, we obtained a model with a size of only 732 KB, and the overall pruning rate of the pruning algorithm for the YOLOXs model was close to 0.9. The number of filters was reduced from 11,570 to 1213. A comparison of the filter changes before and after pruning is shown in Table 1. It is clear that the pruning algorithm effectively reduces the number of filters.

## 4. Results and Discussion

### 4.1. Evaluation Criteria

In deep learning target detection, the *F*1 score is a metric for the classification problem. It is the summed average of the precision and the recall, with a maximum of 1 and a minimum of 0. The *F*1 score is obtained from Equation (14).
(14)$$F1=2\times \frac{P\cdot R}{P+R}$$

In Equation (14), *P* is precision, which represents the proportion of examples identified as positive that are actually positive; *R* is recall, which represents the proportion of the total number of actual positive cases that are correctly identified as positive, and *P* and *R* are derived from the following equations, respectively.
(15)$$P=\frac{TP}{TP+FP}\times 100\%$$
(16)$$R=\frac{TP}{TP+FN}\times 100\%$$

In Equations (15) and (16), *TP* is true positive, *FP* is false positive, and *FN* is false negative.

### 4.2. Results

#### 4.2.1. Wire Break Detection Results of the Well-Tuned YOLOXs Model

Using the well-tuned YOLOXs model for video detection in the test set, the recall, precision, *F*1 score, and false detection rate of the model reach 100%, 100%, 1, and 0%, respectively. Some of the detection results are shown in Figure 11, and it is clear that the broken wire event is accurately monitored. The video detection frame rate reached 30 frames to meet the real-time monitoring requirements.

#### 4.2.2. Wire Break Detection Results for the Pruned YOLOXs Model

The iteratively pruned and retrained YOLOXs model is used for the test set video detection. The recall, precision, *F*1 score, and false detection rate of the pruned model reached 100%, 100%, 1, and 0%, respectively; the video detection frame rate reached 32 frames. The results show that the frame rate was improved by two frames without loss of detection precision, and some of the detection results are shown in Figure 12.

### 4.3. Discussion

#### 4.3.1. Comparison before and after Pruning

After pruning, the percentage of parameters in each layer decreases along with the reduction in the percentage of filters, and the comparison of the percentage of filters and parameters before and after pruning is shown in Figure 13. It can be seen that the rate of reduction in the percentage of parameters is much greater than the rate of reduction in the percentage of filters, and that filter pruning provides a good model simplification effect.

To verify the effectiveness of the pruning algorithm for PCCP wire break detection, the detection results of YOLOXs before and after pruning are compared. The comparison results are summarized in Table 2. It can be seen that the number of parameters is reduced by 98.45%, the number of filters is reduced by 89.52%, the number of inference frames is increased by 2, and the model size is only 732 KB, which is 2.1% of that of the original model. Meanwhile, the *F*1 scores were both 1, with no changes in detection precision. The comparison results show that the pruning algorithm greatly reduces the number of parameters, filters, and model size while guaranteeing the detection precision, and the inference speed increases slightly.

#### 4.3.2. Fusing BN and Convolution Layers to Accelerate Inference

After fusing the BN and convolution layers, the pruned YOLOXs model is used for detection; as the fusion process is an equivalent replacement, the detection precision does not change but the inference time is reduced. Under the video test, the detection frame rate is further increased to 35 fps and some of the detection results are shown in Figure 14.

## 5. Conclusions

In this study, deep learning techniques are applied to the monitoring of wire breaks in PCCPs. The pruning algorithm is used in a YOLOXs model, and spectrograms obtained by synchrosqueezed wavelet transform of the acoustic signal are taken to train the network for the detection of wire break events in the PCCP. After pruning, the detection precision remained unchanged, the number of model parameters was reduced by 98.45%, and the model size was only 732 KB, being only 2.1% of the size before pruning. The experimental result shows that the method greatly simplifies the model, with guaranteed precision. Its recall, precision, *F*1 score, and false detection rate reach 100%, 100%, 1, and 0%, respectively; after accelerating the model inference by fusing the convolution and BN layers, the detection frame rate reaches 35 fps when detecting a 24 fps 480p video, which can meet the demand of the real-time monitoring. At the same time, the lightweight model can be better deployed in the PCCP wire break monitoring system.

In the future, the research team will continue to focus on the following research directions. Firstly, we will set up the DAS equipment in actual operating PCCP pipelines, completing actual monitoring trials and trying to obtain real wire break data. Secondly, we wish to improve the accuracy of wire break location detection and to determine the warning threshold for PCCP pipe bursts in conjunction with other research. Finally, we aim to complete the development of a PCCP broken wire burst warning system.

## Figures and Tables

**Figure 1 sensors-23-02090-f001:**
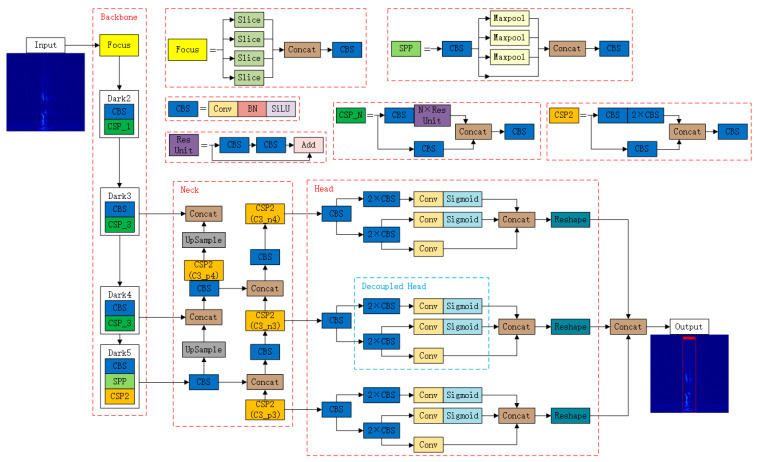
Spectrogram detection of wire breaks based on YOLOXs.

**Figure 2 sensors-23-02090-f002:**
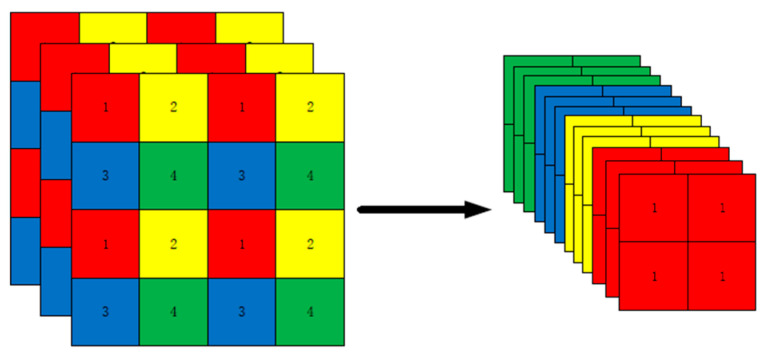
Image slicing by the Focus module.

**Figure 3 sensors-23-02090-f003:**
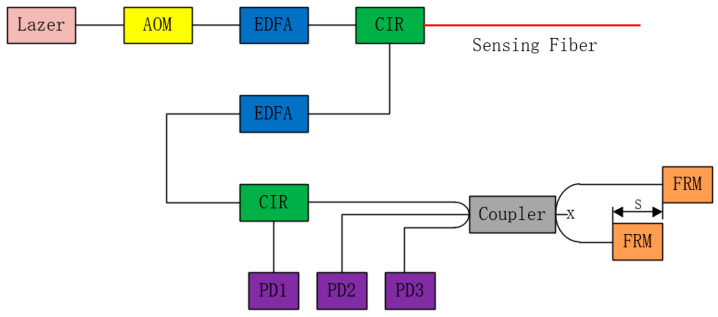
The experimental setup of the DAS system.

**Figure 4 sensors-23-02090-f004:**
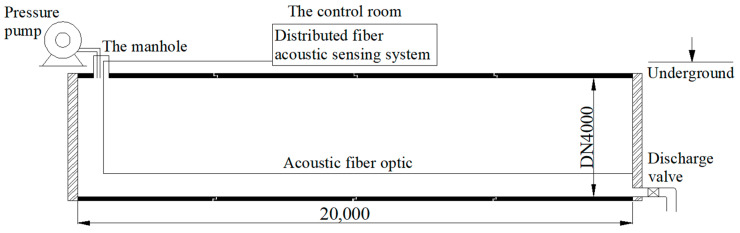
Sketch of the test setup.

**Figure 5 sensors-23-02090-f005:**
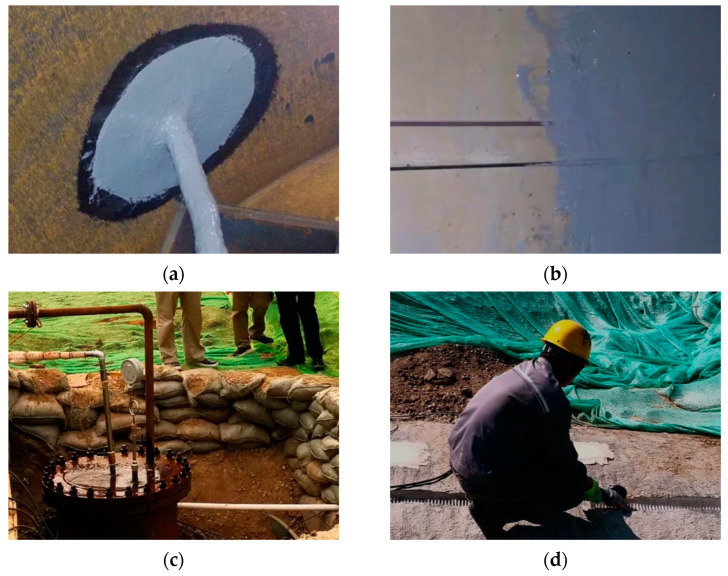
Test procedure: (**a**) inlet cable sealing, (**b**) cable distribution, (**c**) pressurization, (**d**) wire cutting.

**Figure 6 sensors-23-02090-f006:**
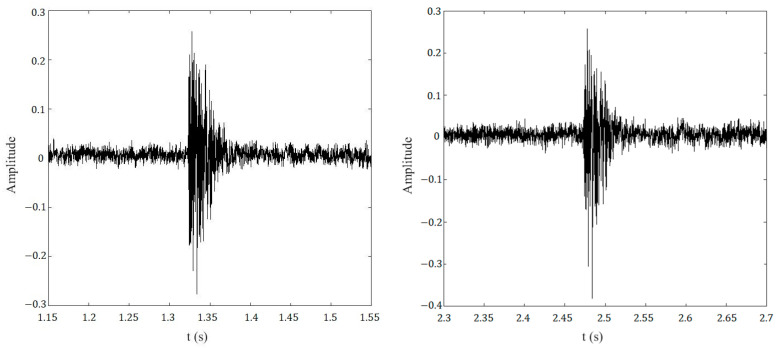
Typical wire break acoustic signal waveforms.

**Figure 7 sensors-23-02090-f007:**
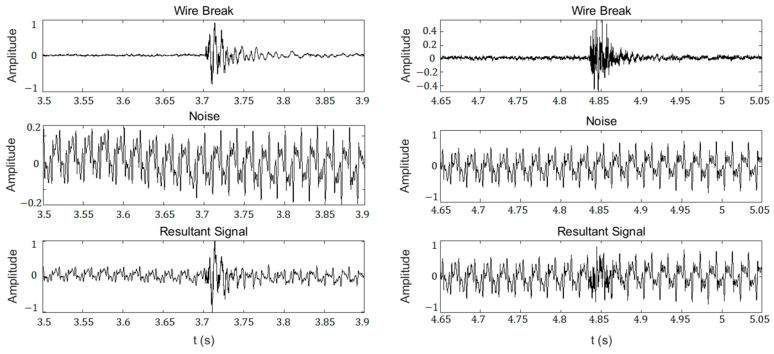
Typical acoustic signal combination processes.

**Figure 8 sensors-23-02090-f008:**
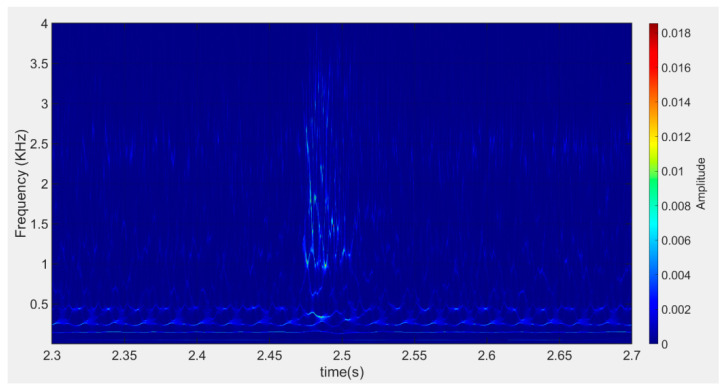
Typical spectrogram transformed by SWT.

**Figure 9 sensors-23-02090-f009:**
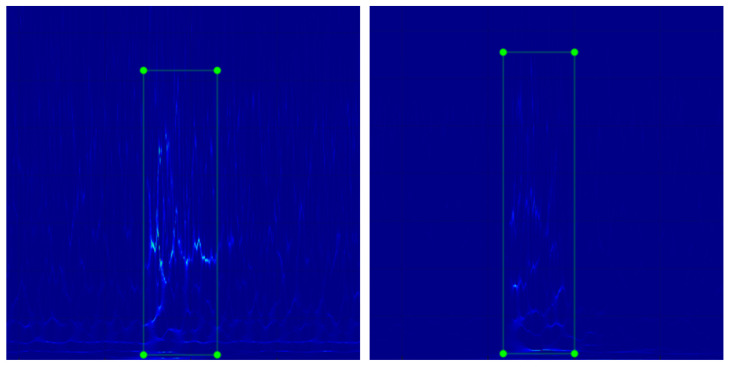
Wire break spectrogram dataset.

**Figure 10 sensors-23-02090-f010:**
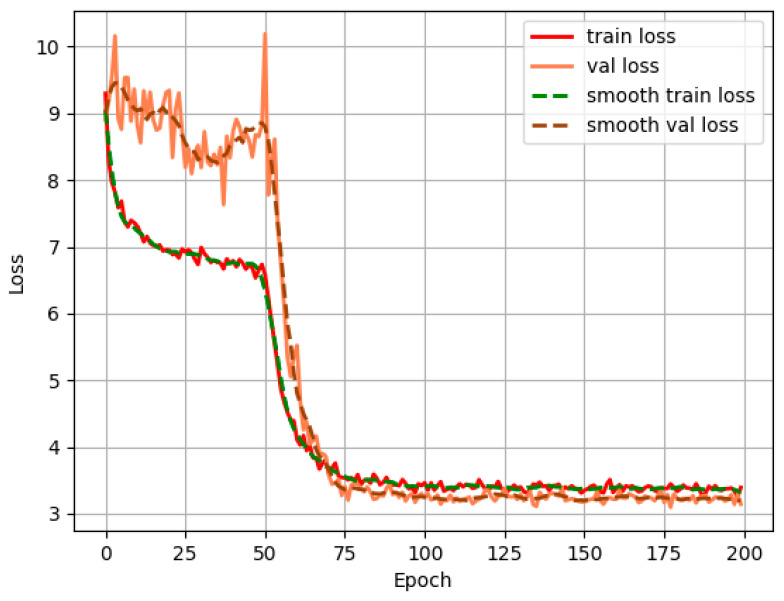
Scaling curve of network loss function.

**Figure 11 sensors-23-02090-f011:**
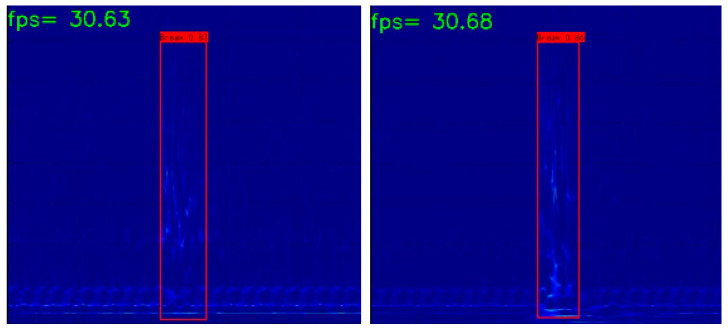
Examples of wire break detection results based on well-tuned YOLOXs.

**Figure 12 sensors-23-02090-f012:**
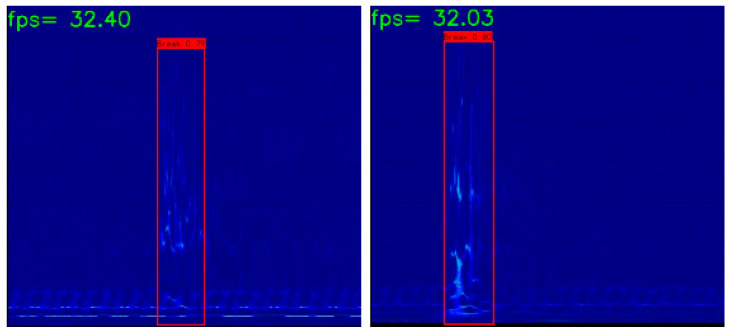
Examples of wire break detection results based on pruned YOLOXs.

**Figure 13 sensors-23-02090-f013:**
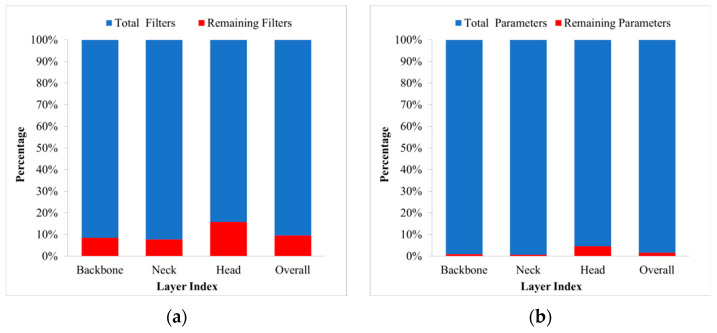
Changes in filters and parameters before and after pruning: (**a**) change in filters, (**b**) change in parameters.

**Figure 14 sensors-23-02090-f014:**
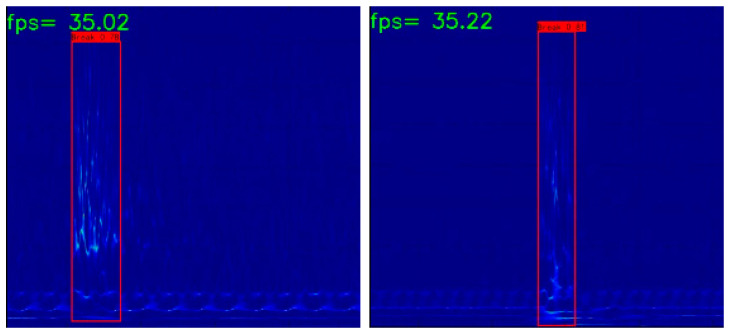
Examples of wire break detection results after fusing.

**Table 1 sensors-23-02090-t001:** Change in filters before and after pruning.

Layer	Backbone		Neck		Head		Overall	
Model	YOLOXs	Pruned YOLOXs	YOLOXs	Pruned YOLOXs	YOLOXs	Pruned YOLOXs	YOLOXs	Pruned YOLOXs
Number of Filters	5408	502	4224	348	1938	363	11,570	1213
Pruning Rate	0.91		0.92		0.81		0.90	

**Table 2 sensors-23-02090-t002:** Comparison of the parameters before and after pruning.

	Number of Parameters	Number of Filters	Model Size	*F*1 Score	Inference
YOLOXs	8,619,648	11,570	34.3 MB	1	30 fps
Pruned YOLOXs	133,629	1213	732 KB	1	32 fps

## Data Availability

Not applicable.

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
