# Peer review of "A YOLOX-Based Automatic Monitoring Approach of Broken Wires in Prestressed Concrete Cylinder Pipe Using Fiber-Optic Distributed Acoustic Sensors"

_sensors, 2023, doi:10.3390/s23042090_

Round 1
Reviewer 1 Report
The introduction is clear, but could be a little shorter. At present it is rather dominant.
Author Response
Response to Reviewer 1 Comments
Point 1: The introduction is clear, but could be a little shorter. At present it is rather dominant.
Response 1: Thank you for the positive statements and the kindly suggestion. The authors have tried to reduce the content of the introduction appropriately in the revised manuscript.

Reviewer 2 Report
This manuscript introduces a YOLOX-Based target detection algorithm to detect the occurrence of wire breakage events by extracting the spectrogram image features of wire break signals in the dataset. In summary, the research is interesting and provides valuable results, but the current document has several weaknesses that must be strengthened to obtain a documentary result that is equal to the value of the publication.
(1) At the thematic level, the proposal provides a very interesting vision, as the automation of wire breakage events detection in PCCP would be a very useful resource for engineers. Furthermore, the detection algorithm model is improved to achieve good accuracy and speed. From the perspective of engineering application, the lightweight model is more conducive to the embedded deployment of the PCCP wire break monitoring system.
(2) The document contains a total of 51 employed references, of which 27 are publications produced in the last 5 years (53%). In this way, the total number is sufficient, and their actuality is high. Moreover, the authors are encouraged to present a broad range of structural health monitoring technologies briely. For example, computer vision detection system (novel visual crack width measurement based on backbone double-scale features for improved detection automation) and Acoustic emission monitor.
(3) The algorithm model has not been verified in the actual engineering environment. Furthermore, has the author considered the maximum length of PCCP that can be detected by this algorithm?
(4) In chapter 1 Introduction, the author summarized the Automatic Monitoring Approach of Broken Wires in PCCP, in which, the development of key sensor technologies related to the topic and the application of the latest deep learning technology are introduced, and the main work of this research is presented at the end.
(5) Vision technology integrated with deep learning is emerging these years in various engineering fields. The authors may add more state-of-art articles for the integrity of the introduction. For visual measurement, please refer to Seismic performance evaluation of recycled aggregate concrete-filled steel tubular columns with field strain detected via a novel mark-free vision method.
(6) In chapter 2, the author makes the theoretical derivation of signal data processing and algorithm model improvement, to meet the requirements of this study on algorithm model detection accuracy and detection speed.
(7) Chapter 3.1. Neural Network Architecture: The subtitle of this section should be replaced with "2.2".
(8) Chapter 3.2.3 Fine Tune YOLOXs: in the Scaling curve of the network loss function, the "Val loss" has shown a tendency to rise sharply and then fall again. Please give a reasonable explanation or adjust the data set and learning rate.
(9) Figure 13 should be modified as Figure 12.
(10) It should mention the scope for further research as well as the implications/application of the study.
(11) I recommend including the limitations regarding the consideration of damage indicated in this review in the limitations assessment. This part of the document can be improved and completed with more rigour.

Author Response
Response to Reviewer 2 Comments
Point 1: At the thematic level, the proposal provides a very interesting vision, as the automation of wire breakage events detection in PCCP would be a very useful resource for engineers. Furthermore, the detection algorithm model is improved to achieve good accuracy and speed. From the perspective of engineering application, the lightweight model is more conducive to the embedded deployment of the PCCP wire break monitoring system.
Response 1: Thanks for your comment.
Point 2: The document contains a total of 51 employed references, of which 27 are publications produced in the last 5 years (53%). In this way, the total number is sufficient, and their actuality is high. Moreover, the authors are encouraged to present a broad range of structural health monitoring technologies briefly. For example, computer vision detection system (novel visual crack width measurement based on backbone double-scale features for improved detection automation) and Acoustic emission monitor.
Response 2: Thanks for your positive evaluation and the valuable comment. Some structural health monitoring techniques have been presented in instruction.
Point 3: The algorithm model has not been verified in the actual engineering environment. Furthermore, has the author considered the maximum length of PCCP that can be detected by this algorithm?
Response 3: Thanks for your comment. The algorithm has not yet been applied in practical engineering and, in addition, the maximum length of PCCP that can be detected by the algorithm is related to the detection accuracy of the DAS instrument and the length that it can detect. Improving the detection accuracy and length of DAS instruments would be another area of scientific interest.
Point 4: The document contains a total of 51 employed references, of which 27 are publications produced in the last 5 years (53%). In this way, the total number is sufficient, and their actuality is high. Moreover, the authors are encouraged to present a broad range of structural health monitoring technologies briefly. For example, computer vision detection system (novel visual crack width measurement based on backbone double-scale features for improved detection automation) and Acoustic emission monitor.
Response 4: Thanks for your comment.
Point 5: Vision technology integrated with deep learning is emerging these years in various engineering fields. The authors may add more state-of-art articles for the integrity of the introduction. For visual measurement, please refer to Seismic performance evaluation of recycled aggregate concrete-filled steel tubular columns with field strain detected via a novel mark-free vision method.
Response 5: Thanks for your comment. Some Vision technologies integrated with deep learning have been presented in instruction.
Point 6: In chapter 2, the author makes the theoretical derivation of signal data processing and algorithm model improvement, to meet the requirements of this study on algorithm model detection accuracy and detection speed.
Response 6: Thanks for your comment.
Point 7: Chapter 3.1. Neural Network Architecture: The subtitle of this section should be replaced with "2.2".
Response 7: Thanks for your comment. This problem has been fixed.
Point 8: Chapter 3.2.3 Fine Tune YOLOXs: in the Scaling curve of the network loss function, the "Val loss" has shown a tendency to rise sharply and then fall again. Please give a reasonable explanation or adjust the data set and learning rate.
Response 8: Thanks for your comment. The problem of the "val loss" showing a sharp increase and then a decrease is mainly due to the fact that the value of the batch size was set to 8, a relatively small value, which led to a significant oscillation of the "val loss" in the first 50 epochs, while increasing the batch size also created new problems, for example, leading to a decrease in the generalisation ability of the model. This problem was also noted during training, but from the 50th epochs, the "val loss" rapidly decreased and converged, indicating that the parameters were chosen to be feasible.
Point 9: Figure 13 should be modified as Figure 12.
Response 9: Thanks for your comment. This problem has been fixed.
Point 10: It should mention the scope for further research as well as the implications/application of the study.
Response 10: Thanks for your comment. The scope of further research is included in the conclusions now.
Point 11: I recommend including the limitations regarding the consideration of damage indicated in this review in the limitations assessment. This part of the document can be improved and completed with more rigour.
Response 11: Thanks for your comment. This part of the document has been improved now.

Reviewer 3 Report
1. (Line 341) Please explain the influence of fiber optic cable packaging method on actual measurement results. Also the main novelty and advantages of the method should also be
2. (Line 350) Whether it is possible to add a variety of different external forces to cut the prestressed wire test and compare?
3. (Line 377) This pipeline background noise data is collected under what kind of external noise environment the pipeline is in. Is there any noise such as human walking and vehicle driving near the pipeline?
4. (Line 386) Although it is mentioned that the generated dataset image only retains the middle part of the image containing break features, and the image size is uniformly scaled to 640 * 640 pixels, can you optimize the image quality to make the color changes in the changed areas clearer? In addition, can the meanings represented by different colors in dataset images be quantified?
5. Figures 7, 9 and 10 do not have abscissa and ordinate. Can you add some explanations?
6. The actual pipeline landfill location is much more complex than the experimental test. Whether the external temperature changes, such as the thermal expansion and cold contraction effects, can be reflected in the wire break detection results based on the well-tuned YOLOXs.
7. Furthermore, the introduction provide sufficient background and include all relevant references? .There are also some typo-mistakes and expression mistakes should be corrected, some figures are with the wrong numbers in the manuscript.

Author Response
Response to Reviewer 3 Comments
Point 1: (Line 341) Please explain the influence of fiber optic cable packaging method on actual measurement results. Also the main novelty and advantages of the method should also be
Response 1: Thanks for your comment. Optical cable packaging is a necessary measure to protect optical cables from being damaged during installation and operation, and has no effect on the actual measurement results. The main innovation of this paper is the application of deep learning techniques to the identification of PCCP broken wire signals obtained from tests. This study improves an algorithmic model to make it suitable for application in a PCCP broken wire monitoring system for continuous monitoring of PCCP broken wires. The advantages of the model include automated identification, fast detection, high accuracy and small model size.
Point 2: (Line 350) Whether it is possible to add a variety of different external forces to cut the prestressed wire test and compare?
Response 2: Thanks for your comment. The main causes of wire fracture are chemical corrosion, electrochemical corrosion and hydrogen embrittlement. Wire breakage due to hydrogen embrittlement is not capable of being simulated by our research team. Chemical and electrochemical corrosion wire breakage is the same as cutting wire breakage, both essentially reduce the cross-sectional area of the wire leading to wire breakage and their acoustic characteristics are similar. Whereas simulated corrosion wire breakage can consume a large amount of time and the breakage time cannot be precisely controlled, this test uses a pressurised test rig in a PCCP pipe plant, and too long an occupation would interfere with the plant's production activities.
Point 3: (Line 377) This pipeline background noise data is collected under what kind of external noise environment the pipeline is in. Is there any noise such as human walking and vehicle driving near the pipeline?
Response 3: Thanks for your comment. The pipeline background noise data was collected in a pipeline buried to a depth of 3m in an urban environment with human activity and vehicle movements, and the presentation on background noise has been clarified in the revised manuscript.
Point 4: (Line 386) Although it is mentioned that the generated dataset image only retains the middle part of the image containing break features, and the image size is uniformly scaled to 640 * 640 pixels, can you optimize the image quality to make the color changes in the changed areas clearer? In addition, can the meanings represented by different colors in dataset images be quantified?
Response 4: Thank you for the comment and the kindly suggestion. Apologies, the scale was an inaccurate representation, actually the 640*640 pixel image was obtained by cutting, the image quality was not changed, this description has been corrected in the revised manuscript. A description image with the meaning of the horizontal and vertical coordinates of the spectrograms and what the colors represent was added in figure 8 in section 3.2.2.
Point 5: Figures 7, 9 and 10 do not have abscissa and ordinate. Can you add some explanations?
Response 5: Thanks for your comment. An illustration of the meaning of the horizontal and vertical coordinates and colors of the spectrogram is added to section 3.2.2. They will not be repeated in Figures 7, 9 and 10.
Point 6: The actual pipeline landfill location is much more complex than the experimental test. Whether the external temperature changes, such as the thermal expansion and cold contraction effects, can be reflected in the wire break detection results based on the well-tuned YOLOXs.
Response 6: Thanks for your comment. Microbending losses in optical fibers due to temperature changes are caused by thermal expansion and contraction. The coefficient of thermal expansion of SiO2, which constitutes the optical fiber, is very small and hardly shrinks when the temperature decreases. But the optical fiber in the cable process must be coated and some other components, coated materials and other components of the expansion coefficient is larger, when the temperature decreases, shrinkage is more serious, so when the temperature changes, the material expansion coefficient is different, will make the fiber microbending, especially in the low-temperature area, when the temperature drops to -55°C or so, the additional loss increases sharply, and this temperature is much lower than the buried water pipeline, so the effect is negligible.
Point 7: Furthermore, the introduction provide sufficient background and include all relevant references? .There are also some typo-mistakes and expression mistakes should be corrected, some figures are with the wrong numbers in the manuscript.
Response 7: Thanks for your comment. Some changes to section of the introduction have been clarified in the revised version, while some typos, expression errors and numbering have also been corrected.

Round 2
Reviewer 3 Report
The authors also agree that "Optical cable packaging is a necessary measure to protect optical cables from being damaged during installation and operation",however I cannot agree with the opinion that "Optical cable packaging has no effect on the actual measurement results. " Actually, it is important for actual measurement. Additional experiments are needed for demonstrating the effectiveness of the deep learning technique, thus this paper is not appropriate for publication in Sensor. It is encouraged to publish experimental and theoretical results in as much detail as possible. The full experimental details must be provided so that the results can be reproduced.
Author Response
Response to Reviewer 3 Comments
Point 1: The authors also agree that "Optical cable packaging is a necessary measure to protect optical cables from being damaged during installation and operation", however I cannot agree with the opinion that "Optical cable packaging has no effect on the actual measurement results. " Actually, it is important for actual measurement. Additional experiments are needed for demonstrating the effectiveness of the deep learning technique, thus this paper is not appropriate for publication in Sensor. It is encouraged to publish experimental and theoretical results in as much detail as possible. The full experimental details must be provided so that the results can be reproduced.
Response 1: Thank you for your comments. First of all, armouring is necessary to protect the fiber, and when I propose here that "no effect on the actual measurement results" does not mean that the measurement results of an armoured fiber are exactly the same as those of a bare fiber, but that the armouring does not affect the identification of break events in this test.
We acknowledge that armoured will attenuate the acoustic signal reaching the fiber core, thus reducing the signal-to-noise ratio. Hence, we maintain the same armouring approach in this experiment, in future studies and eventually in engineering applications. In this way, the absolute effect of armoured on the measurement results will be consistent.
Thanks again and a statement on the effect and consistency of armouring has been added to the latest manuscript (line 362) and we will continue to focus on the method and effect of armouring in subsequent studies.
